# Interplay of Walnut Consumption, Changes in Circulating miRNAs and Reduction in LDL-Cholesterol in Elders

**DOI:** 10.3390/nu14071473

**Published:** 2022-04-01

**Authors:** Judit Gil-Zamorano, Montserrat Cofán, María-Carmen López de las Hazas, Tatiana García-Blanco, Almudena García-Ruiz, Mónica Doménech, Mercè Serra-Mir, Irene Roth, Cinta Valls-Pedret, Sujatha Rajaram, Joan Sabaté, Emilio Ros, Alberto Dávalos, Aleix Sala-Vila

**Affiliations:** 1Laboratory of Epigenetics of Lipid Metabolism, Instituto Madrileño de Estudios Avanzados (IMDEA)-Alimentación, CEI UAM + CSIC, 28049 Madrid, Spain; judit.gil@imdea.org (J.G.-Z.); mcarmen.lopez@imdea.org (M.-C.L.d.l.H.); tatiana.306@hotmail.com (T.G.-B.); almudena.garcia@imdea.org (A.G.-R.); 2Lipid Clinic, Endocrinology and Nutrition Service, Institut d’Investigacions Biomèdiques August Pi i Sunyer, Hospital Clínic, 08036 Barcelona, Spain; mcofan@clinic.cat (M.C.); mdomen@clinic.cat (M.D.); serramir@clinic.cat (M.S.-M.); roth@clinic.cat (I.R.); cintavalls@gmail.com (C.V.-P.); eros@clinic.cat (E.R.); 3CIBER Fisiopatología de la Obesidad y Nutrición (CIBEROBN), Instituto de Salud Carlos III, 28029 Madrid, Spain; 4Center for Nutrition, Healthy Lifestyle and Disease Prevention, School of Public Health, Loma Linda University, Loma Linda, CA 92350, USA; srajaram@lllu.edu (S.R.); jsabate@llu.edu (J.S.); 5Cardiovascular Risk and Nutrition, IMIM (Hospital del Mar Medical Research Institute), 08003 Barcelona, Spain

**Keywords:** alpha-linolenic acid, biomarkers, cholesterol, lipid metabolism, nuts

## Abstract

The mechanisms underlying the lipid-lowering effect of nuts remain elusive. This study explores whether one-year supplementation with walnuts decreases LDL-cholesterol (LDL-C) by affecting the expression of circulating microRNAs (c-miRNA). In this sub-study of the Walnuts and Healthy Aging (WAHA) trial, we obtained fasting serum at baseline and at 1 year from 330 free-living participants (63–79 year, 68% women), allocated into a control group (CG, abstinence from walnuts, *n* = 164) and a walnut group (WG, 15% of daily energy as walnuts, ~30–60 g/d, *n* = 166). Participants in the WG showed a 1 year decrease in LDL-C (−9.07, (95% confidence interval: −12.87; −5.73) mg/dL; *p* = 0.010 versus changes in the CG). We conducted a miRNA array in eight randomly selected participants in the WG who decreased in LDL-C. This yielded 53 c-miRNAs with statistically significant changes, 27 of which survived the correction for multiple testing. When validating them in the full population, statistical significance lasted for hsa-miR-551a, being upregulated in the WG. In mediation analysis, the change in hsa-miR-551a was unrelated to LDL-C decrease. Long-term supplementation with walnuts decreased LDL-C independently of the changes in c-miRNA. The hsa-miR-551a upregulation, which has been linked to a reduced cell migration and invasion in several carcinomas, suggests a novel mechanism of walnuts in cancer risk.

## 1. Introduction

Evidence from randomized controlled trials indicates a cholesterol-lowering effect of nut-supplemented diets [1], particularly among those with higher low-density lipoprotein-cholesterol (LDL-C) [2]. Nuts supply unsaturated fatty acids, the intake of which has been suggested as the main contributor to the reduction in LDL-C ascribed to nut consumption. However, it has been reported that the cholesterol-lowering effect associated with nut supplementation is larger than expected based on predictive equations considering the exchange of fatty acids and cholesterol [3]. Indeed, nuts contain other bioactive compounds that might contribute to both cholesterol reduction and cardiovascular protection [4]. The molecular mechanisms by which nuts improve the lipid profile remain to be fully elucidated.

MicroRNAs (miRNAs) are small noncoding RNAs that negatively regulate gene expression at the post-transcriptional level by generally binding to the 3’ untranslated region of their target messenger RNAs [5]. miRNAs circulate in extracellular fluids, such as plasma, to mediate intercellular communications, being transported in different vesicles (i.e., exosomes, apoptotic bodies, and lipoprotein complexes) [6]. This fostered interest in circulating miRNAs (c-miRNAs) as novel players in lipid metabolism and cardiovascular disease [7,8,9] as well as biomarkers of human diseases such as Huntington’s disease [10]. Evidence has accumulated that c-miRNAs can be modulated by dietary factors, particularly by antioxidants of plant origin [11,12,13]. In line with this, the consumption of nuts, an important source of antioxidants, has been proven to modulate c-miRNAs [14,15,16]. However, the interplay between nut consumption, changes in c-miRNAs, and LDL-C reduction remains to be explored.

The WAHA study (Walnuts and Healthy Aging) is a randomized clinical trial conducted in free-living elders aimed at evaluating the effects of walnut consumption in cognitively healthy elders (63–79 years-old). We already reported that compared to the control group (i.e., usual diet with abstention from walnuts), consumption of walnuts for 2 years improved fasting lipids [17]. Here, we aimed at providing further mechanistic insight into these results, in particular, the reduction of LDL-C. Given that one-year changes in c-miRNAs in a selected subset is a secondary prespecified WAHA outcome [18], we hypothesized that the expected LDL-C reduction after one year of walnut consumption would be partly mediated by changes in c-miRNAs.

## 2. Materials and Methods

### 2.1. Study Design and Participants

The WAHA study is a two-year parallel-group, observer-blinded, randomized controlled trial examining the effects of a diet enriched with walnuts at 15% of energy versus a diet without walnuts in cognitively healthy elderly men and women (https://clinicaltrials.gov/show/NCT01634841, accessed on 20 February 2022). The study was conducted in two centers: Loma Linda University, California, USA (Loma Linda), and Hospital Clínic, Barcelona, Spain (Barcelona). Detailed information on exclusion criteria, assessment of risk factors, randomization and masking, and procedures can be found elsewhere [17]. Briefly, we randomly assigned participants to either the walnut group (consuming ≈15% of daily energy intake as walnuts on top of their habitual diet, ranging from 30 to 60 g of walnuts/day) or the control group (following their usual diet, with abstention from walnuts and avoiding other nuts at doses > 2 serving/week for the duration of the study). The study was conducted in accordance with the guidelines of the Declaration of Helsinki and was approved by the ethics committee of each center. All participants gave written informed consent prior to enrollment. The present sub-study was conducted only with participants completing the first year of intervention at the Barcelona site. In brief, we screened c-miRNA profiles in serum samples (before and after intervention) of 8 randomly selected participants from the walnut arm, and we validated in the whole study population those c-miRNA surviving Benjamini–Hochberg correction for the high false discovery rate (Figure 1).

### 2.2. Laboratory Determinations

At baseline and after 1 year of intervention, fasting blood samples were drawn. Blood samples were centrifuged and aliquoted and stored immediately at −80 °C until analyses. Serum lipid concentrations were determined by standard methods in the hospital clinical laboratory. To objectively determine adherence to supplemental walnuts, in a random sub-sample of 140 participants (*n* = 66 in the control group and *n* = 74 in the walnut group), we measured changes in the red blood cell proportions of alpha-linolenic acid (C18:3 n-3, ALA—an integral compound of walnuts) as described [19].

Screening of c-miRNA profiles was performed in 200 μL of serum samples (before and after intervention) of 8 randomly selected participants from the walnut arm showing any LDL-c reduction after intervention. We extracted c-miRNAs using the miRCURYTM RNA isolation kit, Biofluids (Exiqon, Vedbaek, Denmark), according to the manufacturer’s instructions. The. RNA spike-in kit (Exiqon) was used in all extractions (UniSp2, UniSp4, UniSp5, UniSp6, and cel-miR-39-3p). Next, cDNA was synthesized using the miRCURY LNATM Universal RT miRNA PCR kit (Exiqon) following the manufacturer’s instructions. Then, c-miRNAs were detected by quantitative real-time PCR (qRT-PCR) using miRCURY LNA miRNA miRNome PCR Human panel I+II, V4 (which collectively consist of 752 pre-selected human mature miRNAs) and the ExiLENT SYBR green master mix (Exiqon) on a 7900HT fast Real-Time PCR System (Applied Biosystems, Foster City, CA, USA). Ct values were normalized using spike-in UniSp2, UniSp4, and cel-miR-39-3p, and the data analysis to obtain the c-miRNA expression was performed using GenEx software (MultiD Analyses AB, Göteborg, Sweden).

C-miRNA candidates were then validated in the whole population. To this end, c-miRNAs and cDNA from baseline and 1 year plasma samples were obtained as previously described. Validation was performed by qRT-PCR using the ExiLENT SYBR green master mix (Exiqon) and Pick-&-Mix microRNA PCR Panels, 384-well Ready-to-Use, V4, pre-designed with LNA™ primer sets (Exiqon). The relative c-miRNA expression was calculated using the 2^−ΔΔCt^ method [20].

### 2.3. Nutritional Examination

We scheduled participants for a visit with the dietitians every 2 months to assess compliance, increasing retention, collecting data on tolerance, and delivering walnuts when appropriate. Food consumption was monitored through 3 day food records at both baseline and 1 y visits, and we calculated the nutrient composition of the diets with Food Processor Plus software (ESHA Research, Salem, OR, USA), adapted to nutrient databases of local foods when appropriate.

### 2.4. Statistical Analyses

This study was initially conceived as an opportunistic WAHA sub-study and, therefore, no specific power calculation was conducted. Normal distribution of data was assessed using graphical methods and the Shapiro–Wilk test. Categorical variables are expressed as frequencies and percentages. Quantitative variables following a normal distribution are expressed as means (95% confidence intervals).

We assessed between-group differences in baseline characteristics of the study population by the chi-square test, one-way analysis of variance, and the Kruskal–Wallis test, as appropriate. We assessed between-group changes in energy and nutrient intake as well as the ALA proportion of red blood cell membranes by one-way analysis of variance. Changes in fasting serum lipids were further assessed by analysis of covariance adjusted by gender, age, baseline values of each dependent variable, in-trial change in self-reported physical activity (MET-min/day), remaining statin-naive during the intervention (yes/no), and in-trial change in statin doses standardized to simvastatin.

We assessed differences at baseline between participants undergoing c-miRNA screening and those not screened by the chi-square test, one-way analysis of variance, or the Kruskal–Wallis test as appropriate. Regarding c-miRNA screening in a selected sub-sample of participants, after pre-processing of raw data and normalization using exogenous RNAs (spike in), we assessed the differences between baseline and 1 year c-miRNA expression by paired *t*-tests using Benjamini–Hochberg correction for the high false discovery rate (FDR). We then validated for the whole study population the effect of the intervention on selected c-miRNAs by using repeated-measures analysis of covariance with 2 factors: time (baseline vs. 1 year), group (control vs. walnuts), and their interactions with age, gender, remaining statin-naive during the intervention (yes/no) and in-trial change in statin doses standardized to simvastatin as covariates.

Last, we conducted a mediation model in the whole study population, considering as mediators those c-miRNAs in which the levels changed significantly in the whole population after the dietary intervention. This model sought to further explore the observed relationship between the independent variable (dietary intervention) and the dependent one (change in LDL-C, using standardized residuals outputted from general linear models including age, gender, baseline LDL-C, remaining statin-naive during the intervention (yes/no), and in-trial change in statin doses (standardized to simvastatin) via the inclusion of a third hypothetical variable (change in c-miRNA), known as the mediator. Different paths were created in this model: Path a, representing the effect of dietary intervention on a selected c-miRNA; path b, representing the effect of change of a selected c-miRNA on changes in LDL-C; path a × b, known as an indirect effect, which represents the mediated effect of dietary intervention on changes in LDL-C by the mediator; path c, known as a total effect, representing the total effect of dietary intervention on changes in LDL-C; and path c’, known as the direct effect, which represents the remaining effect of dietary intervention on changes in LDL-C not mediated by the variables included in the model.

Analyses were performed using SPSS software, release 19.0 (IBM Corp., Armonk, NY, USA), the GenEx Pro qPCR data analysis software (Exiqon), and R-based open source software Jamovi (The Jamovi project (2019); retrieved from https://jamovi.org (accessed on 20 February 2022)). A *p*-value < 0.05 was considered statistically significant.

## 3. Results

From 642 pre-screened potential candidates at the Barcelona site, 352 participants were finally randomly assigned to either the walnut group (*n* = 179) or the control group (*n* = 173). Three hundred and thirty participants completed the first year of intervention (94% retention rate). A flow chart of the participants throughout the study can be found in Figure 1.

The baseline characteristics of the study’s population by group allocation are presented in Table 1. Red blood cell ALA status (baseline and one-year change) by intervention group are depicted in Figure 2. While there were no significant between-group differences in red blood cell ALA at baseline (mean (95% confidence interval, CI), 0.12 (0.12; 0.13) for the control group and 0.12 (0.12; 0.13) for the walnut group), the one-year change in the walnut group (mean (95% CI), 0.28 (0.24; 0.32)) was significantly higher than that observed in the control group (mean (95% CI), 0.11 (0.08; 0.14)), confirming good adherence to the intervention. Baseline and one-year changes in fasting lipids by intervention group are presented in Table 2. At 1 year, participants in the walnut group significantly decreased total and LDL-C from baseline and when compared to the control group, while no changes were observed in HDL-C or triglycerides.

Concerning c-miRNAs, we first screened changes in a selected group of participants allocated to the walnut arm displaying any reduction of LDL-C at 1 year. Therefore, we excluded those allocated to the walnut arm displaying any LDL-C increase (*n* = 37; mean: 8 mg/dL; range: 1; 15 mg/dL). To limit the influence of factors other than the dietary intervention, we also excluded from being potential participants to be screened those who did not remain statin-naive during the intervention (*n* = 69), smokers (*n* = 6), and those who reported in-trial changes in physical activity > 1500 MET-min/day in absolute value (*n* = 7). This resulted in 45 participants who were suitable for c-miRNA screening. This subgroup showed higher 1 year reductions in total cholesterol (mean: −18; 95% CIs: −26; −13 mg/dL) and LDL-C (mean: −19; 95% CIs: −25; −14 mg/dL) than the rest of the participants randomly allocated into the walnut group (*n* = 121; total cholesterol, mean: −4; 95% CIs: −9; 2 mg/dL; LDL-C, mean: −4; 95% CIs: −8; 1 mg/dL) (*p* < 0.001, both). Eight of the 45 candidates were randomly selected. We found no significant differences between screened (*n* = 8) and non-screened (*n* = 37) participants for demographics and 1 year changes in adiposity and dietary data (Table 3).

For c-miRNA screening, samples were run against human panels containing 752 unique miRNAs probes. Using qRT-PCR, 347 c-miRNAs were detected (Appendix A). Fifty-three miRNAs were modulated after one-year walnut consumption (Appendix A—*p*-value < 0.05), 36 of them being upregulated and 17 downregulated. Considering Benjamini–Hochberg’s correction for multiple testing, 27 c-miRNAs changed significantly, distributed as 17 upregulated and 10 downregulated (Appendix A—Benjamini–Hochberg *p* < 0.05).

Because of the large number of miRNAs evaluated and the type of pre-processing of missing values used by GenEx program (imputation based on group), we manually revised the modulated miRNAs to correct for possible miRNAs favored by data pre-processing. As a result of this process, we finally excluded from validation miR-10a-3p, miR-154-3p, miR-212-3p, miR-21-3p, miR-146b-3p, miR-638, miR-708-3p, miR-374b-3p, miR-191-3p, miR-885-3p, miR-487a-3p, miR-373-5p, miR-409-5p, and miR-218-5p. We also included two c-miRNAs (i.e., miR-192-5p and miR-330-3p) in the validation despite having a Benjamini–Hochberg of *p* > 0.05, given their documented changes upon supplementation with nuts [13,14]. Table 4 displays the effect of the dietary intervention on selected c-miRNAs for the whole study’s population. As observed, statistically significant interaction time × intervention was only observed for hsa-miR-551a (*p* = 0.006), indicative of an upregulating effect ascribed to supplementation with walnuts.

The mediation model considering this c-miRNA is shown in Figure 3. As expected, there was a significant association for path c (total effect, *p* = 0.020). Path a, which represents the association between dietary intervention and changes in hsa-miR-551a, was significant as well (*p* = 0.007). However, no statistical significance was found for path b, which represents the relationship between the change in hsa-miR-551a and the change in LDL-C (*p* = 0.212). As a result, the mediated effect (a × b), representing the extent to which changes in LDL-c secondary to dietary intervention are mediated by changes in hsa-miRNA-199b-5p, was found to not be statistically significant (*p* = 0.258). Accordingly, the bulk of the statistical significance observed for total effect (path c) was driven by the direct effect (path c’, *p* = 0.012).

## 4. Discussion

In this pre-specified sub-study of the WAHA trial aimed at searching for novel evidence of the mechanisms underlying the long-known lipid-lowering effect of walnut consumption, a diet supplemented with walnuts at ≈15% energy over one year resulted in a significant decrease in serum fasting LDL-C and a significant upregulation of circulating hsa-miR-551a. However, in a mediation model, we observed that the effect of walnut consumption on LDL-c was not mediated by the change in this c-miRNA.

Three findings deserve to be highlighted. First, supplementation with walnuts induced a greater reduction in LDL-C (by roughly 7 mg/dL) than the control diet (absence of walnuts), while HDL-C remained essentially unchanged. This is an expected finding, since a 4.3 mg/dL LDL-C reduction was observed in the parent WAHA trial, involving 2 years of intervention and two sites (i.e., Spain and US) [17]. Second, in contrast to previous studies that reported changes for many c-miRNAs after nut supplementation [14,15], we only observed statistically significant differences for a single c-miRNA. This apparent discrepancy might rely on methodological differences among the trials, including the study design, the duration of the intervention, and the study population (i.e., age, associated medical conditions, and concomitant drug treatment). Third, in a heterogeneous older population, we uncovered that compared with participants in the control diet, those consuming walnuts for 1 year significantly increased the serum concentrations of hsa-miR-551a, in which the expression has been related to the delayed progression of several cancers. Of note, experimental studies reported that this miRNA inhibits in vitro gastric carcinoma cell migration and invasion [21], metastatic colonization of the liver by colorectal tumor cells [22], and metastasis in breast carcinomas [23]. Concurring with this notion, demethoxycurcumin was found to inhibit ovarian tumor cell proliferation by upregulating miR-551a expression, reinforcing its growth-inhibitory role [24]. Future research is warranted to elucidate whether (i) serum concentrations of hsa-miR-551a is a good surrogate marker of its status in tissues and (ii) upregulation of hsa-miR-551a secondary to walnut consumption could be mechanistically linked to a lower risk of certain cancers/better prognosis in regular walnut consumers. To date, there is a compelling body of observational evidence of a reduced rate of cancer incidence and mortality associated with nut consumption, albeit research specifically focused on walnuts is still scarce [25].

Finally, we reported that sustained walnut consumption affects both serum LDL-C and c-miRNA status, yet their respective changes were unrelated. We therefore failed to identify a novel mechanism underlying the lipid-lowering effect of walnuts. MiRNAs that regulate LDL-C target LDL receptor expression and are mostly liver based [7]. A limitation of our study was the lack of liver histology, but liver biopsy would have been unethical in the context of our study. In this regard, c-miRNAs might not reflect the status of hepatic miRNA. Therefore, the functional significance of c-miRNA changes upon dietary interventions remains questionable, at least concerning LDL biology.

Our study has other limitations. The parent study was designed to assess 2 year changes in cognitive function and retinal health [15], and our results derive from a pre-specified post hoc analysis. In contrast, the strengths of the present study include the large and heterogeneous population concerning medical conditions and concomitant drug treatment, the length of the intervention, the observer-blinded nature of the randomized controlled trial, and the good compliance with the intervention as attested by changes in red blood cell ALA.

## 5. Conclusions

In this sub-study of the WAHA trial, we found that supplementation with walnuts at 15% of energy requirements (~30–60 g/day) for 1 year was associated with reduced LDL-C in a cohort of cognitively unimpaired elders, an effect that was not mediated by changes in c-miRNA. In addition, the intervention increased the circulating levels of hsa-miR-551a, a miRNA the tissue status of which has been mechanistically linked to a reduced capacity of migration and invasion for certain cancers. Future research is warranted to investigate whether increasing circulating hsa-miR-551a secondary to regular walnut consumption might contribute to slowing down cancer progression.

## Figures and Tables

**Figure 1 nutrients-14-01473-f001:**
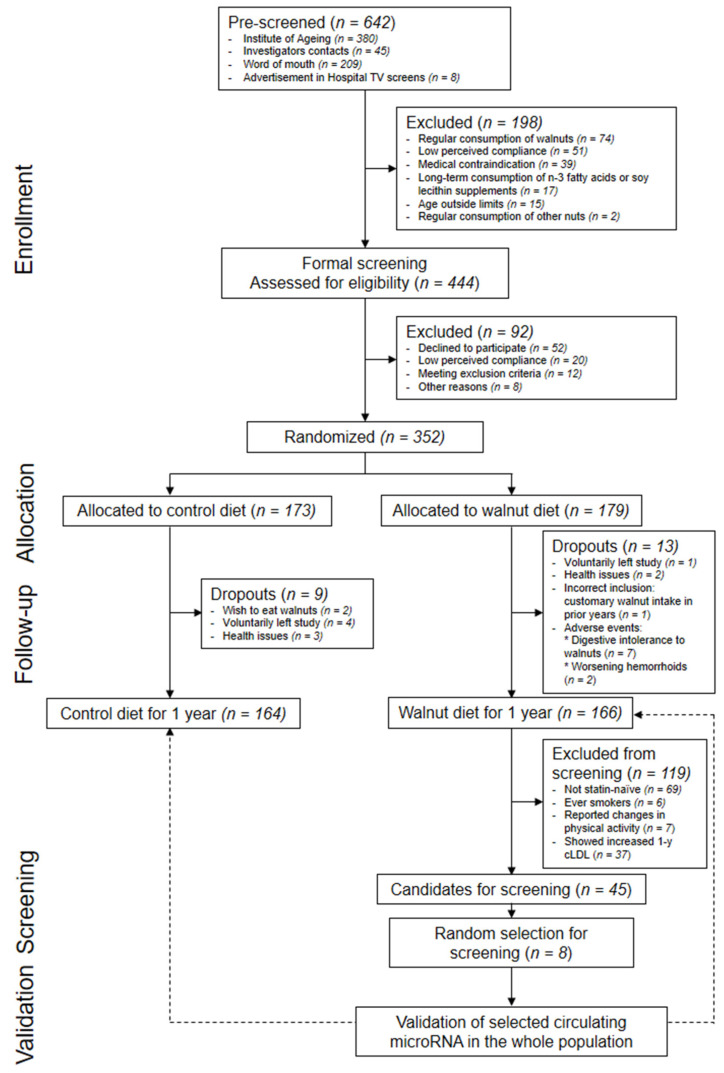
Flow chart of the study.

**Figure 2 nutrients-14-01473-f002:**
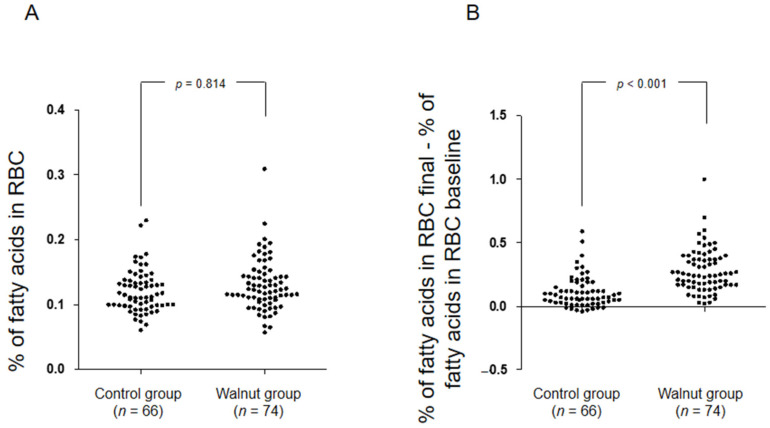
Baseline biomarker of adherence to walnuts and change after one year by intervention. RBCs, red blood cells. (**A**) proportion of alpha-linolenic acid (expressed as percentage of total fatty acids) in RBCs at baseline; *p* obtained by one-way analysis of variance. (**B**) Change in alpha-linolenic acid in RBCs after 1 year, calculated as percent at 1 year minus percent at baseline; *p* obtained by one-way analysis of variance.

**Figure 3 nutrients-14-01473-f003:**
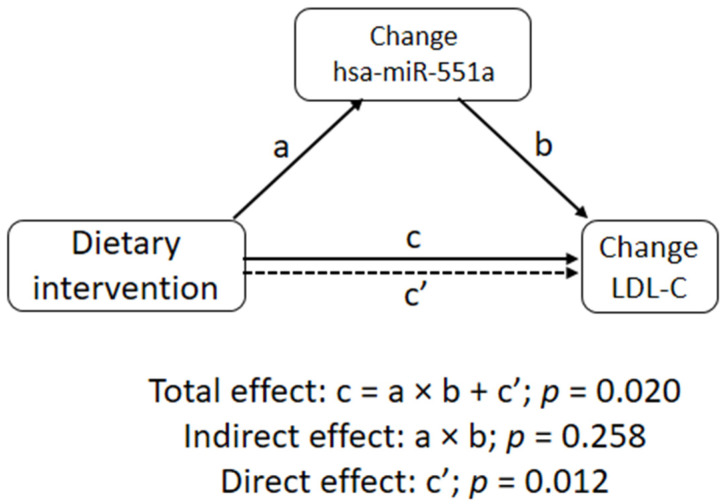
Schematic illustration of the mediation model conducted in the whole study population. Path c (total effect): β = −0.1370, 95% confidence interval (CI) = −0.23735; −0.0201, *p* = 0.020; Path a: β = 0.1581, 95% CI = 0.04389; 0.2787; *p* = 0.007; Path b: β = 0.0742, 95% CI = −0.03902; 0.1756, *p* = 0.212; Path a × b (indirect effect, mediation): β = 0.0117, 95% CI = −0.00806; 0.0301, *p* = 0.258; Path c’ (direct effect): β = −0.1487, 95% CI −0.24925; −0.0302, *p* = 0.012.

**Table 1 nutrients-14-01473-t001:** Baseline characteristics of the study’s population by intervention group.

Variable	Control Diet (*n* = 164)	Walnut Diet (*n* = 166)	*p*-Value ^a^
Women—Number (%)	111 (67.7)	114 (68.7)	0.847
Age—year	68.8 (68.3; 69.3)	69.0 (68.5; 69.5)	0.470
Ever Smoking, Yes—Number (%)	47 (28.7)	46 (27.7)	0.848
Weight—kg	71.0 (69.0; 73.0)	69.9 (68.1; 71.7)	0.405
Body Mass Index—kg/m^2^	27.4 (26.8; 28.1)	26.8 (26.2; 27.3)	0.101
Waist Circumference—cm	99 (97; 101)	97 (96; 99)	0.128
Hypertension—Number (%)	94 (57.3)	89 (53.6)	0.499
Type 2 Diabetes—Number (%)	16 (9.8)	20 (12.0)	0.504
Dyslipidemia—Number (%)	86 (52.4)	89 (53.6)	0.831
Physical Activity ^b^	2412 (1525; 3834)	2629 (1770; 3909)	0.340

Data are *n* (%) or mean (95% confidence interval), except for physical activity, expressed as medians (interquartile ranges). ^a^ Obtained by chi-square test, one-way analysis of variance, or the Kruskal–Wallis as appropriate. ^b^ Physical activity is expressed in MET-min/day, min/day at a given metabolic equivalent level (units of energy expenditure in physical activity, 1 MET-min roughly equivalent to 1 Kcal).

**Table 2 nutrients-14-01473-t002:** Baseline fasting lipids and one-year changes by intervention group.

Variable	Visit	Control Diet (*n* = 164)	Walnut Diet (*n* = 166)	*p*-Value ^a^
Total Cholesterol—mg/dL	Baseline	207.5 (202.5; 212.4)	203.0 (198.0; 207.9)	0.206
	End of intervention	204.8 (199.7; 209.9)	195.1 (189.9; 200.4)	0.010
	Delta	−2.7 (−7.0; 1.6)	−7.9 (−12.1; −3.7)	0.085
	Adjusted change	−1.91 (−5.73; 1.91)	−9.07 (−12.87; −5.28)	0.010
LDL-cholesterol—mg/dL	Baseline	129.0 (124.6; 133.3)	124.8 (120.6; 129.1)	0.179
	End of intervention	126.2 (121.6; 130.7)	117.0 (112.5; 121.6)	0.005
	Delta	−2.81 (−6.91; 1.29)	−7.80 (−11.53; −4.07)	0.076
	Adjusted change	−1.97 (−5.45; 1.51)	−8.86 (−12.31; −5.40)	0.006
HDL-cholesterol—mg/dL	Baseline	58.3 (56.2; 60.5)	57.6 (55.5; 59.8)	0.648
	End of intervention	58.8 (56.6; 61.0)	58.1 (55.9; 60.3)	0.667
	Delta	0.43 (−0.24; 1.10)	0.46 (−0.25; 1.17)	0.960
	Adjusted change	0.41 (−0.29; 1.11)	0.40 (−0.29; 1.09)	0.984
Triglycerides—mg/dL	Baseline	99.9 (93.2; 106.6)	101.8 (95.0; 108.6)	0.693
	End of intervention	99.3 (92.7; 106.6)	101.8 (95.0; 108.6)	0.974
	Delta	−0.63 (−6.63; 5.37)	−2.39 (−7.19; 2.41)	0.650
	Adjusted change	1.20 (−6.14; 3.75)	−2.40 (−7.31; 2.52)	0.736

Data are the means (95% CIs). ^a^ For the baseline, end of intervention, and delta, the values were obtained by one-way analysis of variance. For the adjusted change, the value was obtained by ANCOVA adjusting for age, gender, age, baseline value of each dependent variable, change in self-reported physical activity (MET-min/day), remaining statin-naive (yes/no), and in-trial change in statin doses standardized to simvastatin (*n* = 162 for the control diet and *n* = 164 for the walnut diet, since there were four missing values for changes in self-reported physical activity). LDL-cholesterol; low-density lipoprotein-cholesterol; HDL-cholesterol, high-density lipoprotein-cholesterol.

**Table 3 nutrients-14-01473-t003:** Demographic, adiposity, and dietary data in LDL-c responders after 1 y walnut supplementation by selection to undergo c-miRNA screening array.

Variables	Randomly Selected for c-miRNA Screening (*n* = 8)	Non-Selected (*n* = 37)	*p*-Value ^a^
1 y Change in LDL-cholesterol—mg/dL	−18.8 (−26.9; −10.6)	−19.2 (−25.7; −12.7)	0.954
Women—Number (%)	5 (62.5)	27 (73.0)	0.553
Baseline Age—y	69.8 (66.4; 73.1)	69.0 (67.9; 70.1)	0.581
1 y Change in Weight—kg	0.04 (−0.57; 0.65)	0.33 (−0.41; 1.06)	0.718
1 y Change in Body Mass Index—kg/m^2^	0.01 (−0.22; 0.23)	0.09 (−0.19; 0.38)	0.771
1 y Change in Waist Circumference—cm	0.0 (−0.6; 0.6)	0.1 (−0.9; 1.2)	0.904
1 y Change in Physical Activity ^b^	−56 (−300; 472)	0 (−490; 0)	0.456
1 y Change in Total cholesterol—mg/dL	−15.9 (−22.8; −8.9)	−20.6 (−28.0; −13.1)	0.565
1 y Change in HDL-cholesterol—mg/dL	1.3 (−0.7; 3.2)	−0.3 (−1.8; 1.3)	0.375
1 y Change in Triacylglycerols—mg/dL	8.1 (−8.2; 24.5)	−2.8 (−13.3; 7.7)	0.355
1 y Change in Energy Intake—kcal/day	251 (−90; 593)	136 (13; 258)	0.433
1 y Change in Protein Intake—% Energy	−1.3 (−3.1; 0.4)	−1.5 (−2.7; −0.4)	0.875
1 y Change in Carbohydrate Intake—% Energy	−6.1 (−12.8; 0.6)	−2.4 (−5.0; 0.1)	0.235
1 y Change in Total Fat Intake—% Energy	8.1 (1.5; 14.7)	5.4 (3.3; 7.5)	0.304
SFA—% Energy	0.1 (−2.0; 2.3)	−0.4 (−1.1; 0.4)	0.598
MUFA—% Energy	−2.5 (−6.5; 1.6)	−1.8 (−3.4; −0.2)	0.731
PUFA—% Energy	9.2 (6.6; 11.9)	8.0 (7.0; 8.9)	0.287
ALA—% Energy	2.00 (1.59; 2.41)	1.66 (1.49; 1.83)	0.095
1 y Change in Fiber Intake—g/day	1.63 (−4.20; 7.46)	1.99 (−0.28; 4.27)	0.891
1 y Change in Alcohol Intake—g/day	1.25 (−4.73; 7.22)	−1.29 (−4.02; 1.44)	0.422
1 y Change in Cholesterol Intake—mg/day	−14.5 (−111.4; 82.4)	−22.8 (−59.3; 13.6)	0.847

Data are *n* (%) or mean (95% CIs), except for physical activity, expressed as median (interquartile range). LDL-cholesterol; low-density lipoprotein-cholesterol; HDL-cholesterol, high-density lipoprotein-cholesterol; SFAs, saturated fatty acids; MUFAs, monounsaturated fatty acids; PUFAs, polyunsaturated fatty acids; ALA, alpha-linolenic acid. ^a^ Obtained by one-way analysis of variance, the Kruskal–Wallis test, or the chi-square test as appropriate. ^b^ Physical activity is expressed in MET-min/day, min/day at a given metabolic equivalent level (units of energy expenditure in physical activity, 1 MET-min roughly equivalent to 1 Kcal).

**Table 4 nutrients-14-01473-t004:** Baseline and 1 y serum miRNAs by intervention group.

miRNA (Relative Units)	Visit	Control Diet (*n* = 164)	Walnut Diet (*n* = 166)	*p*-Value ^a^
hsa-miR-224-3p	Baseline	0.384 (0.151; 0.617)	0.593 (0.339; 0.846)	0.128
	1 year	0.253 (−0.038; 0.545)	0.722 (0.405; 1.040)	
hsa-miR-551a	Baseline	0.678 (0.502; 0.855)	0.801 (0.632; 0.970)	0.006
	1 year	0.496 (0.330; 0.662)	0.956 (0.798; 1.115)	
hsa-miR-181c-3p	Baseline	0.016 (0.010; 0.021)	0.024 (0.018; 0.029)	0.272
	1 year	0.019 (0.014; 0.025)	0.022 (0.017; 0.028)	
hsa-miR-1271-5p	Baseline	0.312 (0.167; 0.456)	0.706 (0.560; 0.851)	0.312
	1 year	0.432 (−0.121; 0.985)	1.220 (0.665; 1.776)	
hsa-miR-30d-3p	Baseline	0.339 (0.215; 0.462)	0.567 (0.447; 0.687)	0.557
	1 year	0.408 (0.000; 0.816)	0.810 (0.413; 1.208)	
hsa-miR-493-3p	Baseline	0.290 (−0.155; 0.735)	1.154 (0.237; 0.685)	0.366
	1 year	0.423 (−3.528; 4.375)	3.948 (−0.218; 8.111)	
hsa-miR-589-3p	Baseline	0.284 (−0.089; 0.658)	0.882 (0.526; 1.239)	0.796
	1 year	0.239 (−0.073; 0.551)	0.911 (0.614; 1.209)	
hsa-miR-1227-3p	Baseline	0.153 (−1.633; 1.939)	1.616 (−0.139; 3.370)	0.363
	1 year	0.186 (−0.021; 0.392)	0.474 (0.271; 0.677)	
hsa-miR-181c-5p	Baseline	0.398 (0.222; 0.573)	0.614 (0.447; 0.782)	0.704
	1 year	0.408 (0.214; 0.602)	0.678 (0.493; 0.863)	
hsa-miR-130b-5p	Baseline	0.476 (0.318; 0.634)	0.708 (0.552; 0.865)	0.854
	1 year	0.539 (0.341; 0.738)	0.800 (0.603; 0.996)	
hsa-miR-542-5p	Baseline	0.497 (0.269; 0.724)	0.427 (0.179; 0.674)	0.187
	1 year	0.375 (0.177; 0.573)	0.536 (0.321; 0.752)	
hsa-miR-340-3p	Baseline	0.399 (0.248; 0.550)	0.624 (0.471; 0.778)	0.501
	1 year	0.404 (0.206; 0.601)	0.733 (0.531; 0.934)	
hsa-miR-192-5p	Baseline	1.509 (0.641; 2.376)	2.875 (2.024; 3.727)	0.708
	1 year	1.483 (−0.029; 2.995)	3.298 (1.814; 4.782)	
hsa-miR-330-3p	Baseline	0.313 (0.112; 0.514)	0.807 (0.085; 0.583)	0.175
	1 year	0.334 (0.085; 0.853)	1.006 (0.751; 1.260)	

Data are adjusted means (95% CIs). ^a^ Refers to interaction between time (baseline vs. 1 year) and intervention (control diet vs. walnut diet), obtained by two-way repeated measures analysis of covariance, adjusted for age, gender, remaining statin-naive (yes/no), and in-trial change in statin doses standardized to simvastatin.

## Data Availability

Data described in the article, code book, and analytic code will be made available upon request pending application and approval.

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
