# Peer review of "Interplay of Walnut Consumption, Changes in Circulating miRNAs and Reduction in LDL-Cholesterol in Elders"

_nutrients, 2022, doi:10.3390/nu14071473_

Round 1
Reviewer 1 Report
The manuscript by Judit Gil-Zamorano et al, entitled “Interplay of walnut consumption, changes in circulating miRNAs and reduction of LDL-cholesterol in elders” has been reviewed.
The authors aimed to analyze in a mediation analysis whether walnut consumption deduces the LDL plasma concentrations and if this effect is mediated by certain miRNAs.
In general, the topic is of interest, the research questions is formulated in a correct way. The manuscript is well written. Nevertheless, there are major issues that have to be addressed.
- My biggest concern is the fact that the c-miRNA analyses were conducted only on 8 subjects. This could easily explain the lack of significance in their medication analysis. Please provide rationality for such a study design.
- As the authors may be aware, one of the biggest problems with nutritional epidemiology is the fact that their lowers effects are rarely observed in such a small sample, even considering the 330 initially enrolled subjects. This could be a greater source of bias for this study, please comment on how did the authors addressed this problem.
- What is the meaning of figure 1. Please elaborate.
- Authors mention that 37 out of the 166 participants with a walnut enriched diet actually increased their LDL plasma concentrations. This is a major point, please further elaborate.
- Table 2: please include values of baseline, 1 year and the delta.
- Table 3: how many subjects include each group?
- Table 3: specify what the numbers represent.
- Table 2 and Table 3. *P-values= it seems that the authors are using the word “interaction” for something else. As it stands, they are not testing interactions. Please correct.
- Please discuss how miRNAs are currently identified, what are the reasons to believe that those miRNAs quantified in this study may be associated with the research question.
- Please provide the code of the analysis to reproduce your findings in an external server i.e. OSF
- Please provide the data in an external server i.e. OSF
- The authors did a good job of stating some limitations and presenting their findings in a sober way.
Author Response
- My biggest concern is the fact that the c-miRNA analyses were conducted only on 8 subjects. This could easily explain the lack of significance in their medication analysis. Please provide rationality for such a study design. As the authors may be aware, one of the biggest problems with nutritional epidemiology is the fact that their lowers effects are rarely observed in such a small sample, even considering the 330 initially enrolled subjects. This could be a greater source of bias for this study, please comment on how did the authors addressed this problem.
It seems that we might failed to clearly explain that, although the screening was conducted in 8 participants, we validated candidate c-miRNAs in the whole cohort (n=330), and, importantly, mediation analysis was also conducted in the whole study population. We tried to make this notion clearer in the revised version of the study. In “Statistical analysis”, please find: “Last, we conducted a mediation model in the whole study population, considering as mediators those c-miRNAs which levels changed significantly in the whole population after the dietary intervention”. In Figure 4 (former Figure 3), please find: “Schematic illustration of the mediation model conducted in the whole study population”.
We hope we can have been able to satisfactorily address your qualm regarding this issue.
--------------
- What is the meaning of figure 1. Please elaborate.
Former Figure 1 no longer appears to the revised version of the manuscript, since it is redundant with detailed information that can be found in Supplementary Table.
--------------
- Authors mention that 37 out of the 166 participants with a walnut enriched diet actually increased their LDL plasma concentrations. This is a major point, please further elaborate.
Participants were free-living subjects on self-selected diets (and in absence of dietary advice), and therefore some changes might be expected as a result of slight modification of dietary habits. Increases of fasting LDL-C observed in participants supplemented with walnuts for 1 year were not clinically meaningful (mean increase for these 37 participants, 8 mg/dl). However, we expanded this notion in the revised version of the manuscript. Please find: Therefore, we excluded those allocated to the walnut arm displaying any LDL-C increase (n = 37; mean: 8 mg/dl; range: 1 to 15 mg/dl).
--------------
- Table 2: please include values of baseline, 1 year and the delta.
This has been corrected.
--------------
- Table 3: how many subjects include each group?
This has been corrected. As stated in a previous point, the validation of candidate c-miRNAs was conducted in the whole study population, this is “Control diet (n = 164)” and “Walnut diet (n = 166)”.
--------------
- Table 3: specify what the numbers represent.
The reviewer is right in the sense that units were missing from the Table, thanks for pointing this out. Quantification of miRNA transcripts implicates data normalization using endogenous or exogenous reference genes for data correction. Data are expressed in relative units after normalization.
--------------
- Table 2 and Table 3. *P-values= it seems that the authors are using the word “interaction” for something else. As it stands, they are not testing interactions. Please correct.
The reviewer is right in the sense that the P value might be misinterpreted as an interaction in Table 2. We corrected this, besides adding P values for between-group comparisons of baseline, 1-year, and deltas.
Regarding Table 3, we are actually testing the interaction. In “Statistical analyses”, please find: “We then validated in the whole study population the effect of the intervention on se-lected c-miRNAs by using repeated-measures analysis of covariance with 2 factors: time (base-line vs. 1 year) as repeated measure, group (control vs. walnuts) and their interactions, with age, gender, remaining statin-naïve during the intervention (yes/no), and in-trial change in statin doses standardized to simvastatin as covariates”.
--------------
- Please discuss how miRNAs are currently identified, what are the reasons to believe that those miRNAs quantified in this study may be associated with the research question.
For screening purposes, it is usual to use a commercial and predesigned PCR primer set for profiling of up to a certain number of miRNAs, which are selected based on in-house analyses of miRNA expression in blood, serum and plasma samples, as well as on available literature. In our particular case, we used miRCURY LNA miRNA miRNome PCR Human panel I+II, which allows to profile up to 752 unique human miRNAs, and includes endogenous and exogenous miRNA controls for normalization of data results.
We tried to include this notion in the revised version of the manuscript. In “Laboratory determinations”, please find: “Then, c-miRNAs were detected by quantitative real-time PCR (qRT-PCR) using miRCURY LNA miRNA miRNome PCR Human panel I+II, V4 (which collectively consist of 752 pre-selected human mature miRNAs)”.
--------------
- Please provide the code of the analysis to reproduce your findings in an external server i.e. OSF. Please provide the data in an external server i.e. OSF.
As stated in the section of “Data Availability Statement”, data described in the article, code book, and analytic code will be made available upon request pending application and approval".
Reviewer 2 Report
The research is novel, and discussed issue is important. However, I have some pieces of advice for Authors to improve the manuscript.
Line 89, please update -80oC and write degrees with proper sign
In Table 1, please add the information on whether there were any differences in characteristic between intervention groups
Line 184: authors carefully screened patients and extracted the results from the group (n=45) without drugs and physical activity changes during the intervention. Could the authors calculate the mean and median of changes in blood parameters in this subgroup?
The authors mentioned information about diet assessment. Why did the authors not adjust the results by changes in dietary habits or did not evaluate the changes in the diet between baseline and end of the study across groups? Please, add to method the information about nutritional examination.
Author Response
- Line 89, please update -80oC and write degrees with proper sign.
This has been corrected.
--------------
- In Table 1, please add the information on whether there were any differences in characteristic between intervention groups.
Revised Table 1 now includes P values for the between-group comparisons. Accordingly, “statistical analyses” has also been modified to include these tests.
--------------
- Line 184: authors carefully screened patients and extracted the results from the group (n=45) without drugs and physical activity changes during the intervention. Could the authors calculate the mean and median of changes in blood parameters in this subgroup?
These results are included in the revised version of the manuscript. Please find: “This resulted in 45 participants suitable for c-miRNA screening. This subgroup showed higher 1-y reductions of total cholesterol (mean: -18; 95% CIs: -26 to -13 mg/dl) and LDL-C (mean: -19; 95% CIs: -25 to -14 mg/dl) than the rest of participants randomly allocated into the walnut group (n = 121; total cholesterol, mean: -4; 95% CIs: -9 to 2 mg/dl; LDL-C, mean: -4; 95% CIs: -8 to 1 mg/dl) (P < 0.001, both). Eight of the 45 candidates were randomly selected (Supplementary Figure 1 and Supplementary Figure 3)”.
--------------
- The authors mentioned information about diet assessment. Why did the authors not adjust the results by changes in dietary habits or did not evaluate the changes in the diet between baseline and end of the study across groups? Please, add to method the information about nutritional examination.
This is a good point. First, we expanded information regarding nutritional examination (Please find new section 2.3. of the revised manuscript). Second, we respectfully disagree regarding the inclusion of dietary items other than the intervention into the models, since it is widely accepted that interventions based on including a food in the diet of free-living subjects on self-selected diets (and in absence of dietary advice) usually leads to a food and nutrient displacement resulting from the intervention itself. For the particular case of the WAHA study, this has been already reported (please see doi: 10.1017/S0007114517001957). The inclusion of changes in dietary habits might result in over-fitted models, so when designing the statistical approach we decided to solely include non-dietary variables.